# The Right to Equal Health: Best Practice Priorities for Māori with Bipolar Disorder from Staff Focus Groups

**DOI:** 10.3390/healthcare12070793

**Published:** 2024-04-06

**Authors:** Tracy Haitana, Mau Te Rangimarie Clark, Marie Crowe, Ruth Cunningham, Richard Porter, Suzanne Pitama, Roger Mulder, Cameron Lacey

**Affiliations:** 1Department of MIHI, University of Otago, Christchurch 8011, New Zealand; 2Department of Psychological Medicine, University of Otago, Christchurch 8011, New Zealand; 3Department of Public Health, University of Otago, Wellington 6021, New Zealand

**Keywords:** bipolar disorder, Māori, best practice, focus groups, health equity

## Abstract

Bipolar disorder (BD) is a serious mental health condition that is clinically complex to monitor and manage. While best practice guidelines exist, they vary internationally lacking consensus. Indigenous peoples, including Māori in New Zealand, experience higher community rates of BD. While New Zealand practice guidelines recommend providing culturally responsive care to Māori, studies show that Māori do not receive best practice. This qualitative study aimed to share the evidence about patterns of health service use and Māori patient experiences with focus group participants involved in the design and delivery of BD services, to discuss and develop guidelines for best practice for Māori with BD and address areas of unmet need. Three focus groups were conducted with 22 participants involved in the delivery of services to Māori with BD across three sites. Willing participants were sent background information and three focus group questions framed to elicit priority solutions to improve clinical, structural and organisational features of mental health service delivery for Māori patients with BD and their whānau (family). The nominal group technique was used to synthesise responses, and then develop a prioritised list of proposed solutions. Results identified system-level changes required at the clinical, structural and organisational levels of healthcare. Findings further evidence the need for healthcare reform in New Zealand, to be responsive to Māori with BD.

## 1. Introduction

Bipolar disorder is a serious lifelong psychiatric illness, which epidemiologically carries the greatest risk of suicide relative to other mental health conditions [1]. While clinical practice guidelines exist to aid in the assessment, identification and treatment of BDs, management of the condition is complex and responding to relapsing symptoms is a likely feature of a patient’s healthcare [1,2]. Guidelines for BD management are developed based on available empirical evidence, as well as the collective opinions of professionals with relevant expertise [2,3]. However, there is a lack of consensus between international guidelines, raising concerns about their validity, and the need for greater consistency in the approach to clinical practice guideline development [3].

Despite the United Nations and a majority of international member states, declaring in 2007 that Indigenous peoples throughout the world have an equal right to the same standards of mental and physical health as other ethnic groups, significant inequities remain [4,5,6]. There is evidence that inequities include higher rates of BD in some Indigenous community samples, including Māori, the Indigenous peoples of New Zealand, yet limited consideration has been given to the role of structural factors in the expression and reduction of those inequities [7,8,9]. While New Zealand clinical guidelines for BD acknowledge the importance of providing culturally appropriate services that consider Māori needs, including reducing barriers to access and discriminatory care, and increasing physical health monitoring, evidence indicates that Māori do not receive best practice let alone culturally responsive care in New Zealand [2,10,11,12]. The call for change to the design and delivery of mental healthcare by redressing the social conditions and colonial institutions that maintain the status quo of Indigenous mental ill health internationally is also widespread [4,13,14,15].

Like many of the world’s Indigenous peoples, Māori have a shared experience of living in marginalised social, cultural and economic conditions relative to the ethnic majority (New Zealand European) population [4,12]. In New Zealand, Western colonialism and imperialism also underpin patterned adverse health inequities, with the health system being a recognised determinant of health through healthcare design and delivery features that maintain inequitable patterns of privilege and disadvantage via differential access to and through quality healthcare [12,16]. The New Zealand health system has a hierarchical structure governed by a central Health Ministry. Health services include primary care by general practice (GP) doctors, community-based specialist services, outpatient and inpatient hospital services, and providers from non-governmental organisations. Mental healthcare typically requires a primary care referral for specialist mental health assessment and intervention and can include periods of inpatient or community-based specialist-level care before discharge to GP management. Treatment for BD is typically delivered from a psychiatric model, prioritising medication-based stabilisation, but can include input from multidisciplinary teams [17].

While Kaupapa Māori service frameworks (based on Māori knowledges and practices to meet Māori needs) exist, and are acknowledged in the latest New Zealand practice guidelines for the treatment of BD, these services are not available in all regions of New Zealand or for all components of a patient’s healthcare [2,18]. Consequently, relying solely on “by Māori for Māori” services to counter the continued impacts of colonisation on Māori health is unrealistic, as well as inconsistent with the legal, ethical and professional responsibilities of New Zealand services to work in ways that deliver health equity for Māori [11,19,20]. Research with healthcare providers indicates their desire for greater knowledge, training, skills and resources to support them in employing best practice approaches that are effective with Indigenous peoples, indicating clear potential to enhance best practice guidelines both in New Zealand and other Indigenous territories [17,21,22,23,24].

This study was the final phase of a broader project, the Māori and Bipolar Disorder Research Project (MBDRP) that involved three phases designed to identify knowledge and prioritise strategies to improve outcomes for Māori with BD in New Zealand. This phase utilised the quantitative and qualitative findings from the two earlier phases of the project and distributed them to focus group participants. Focus groups were then undertaken with staff involved in the design and delivery of healthcare services for Māori patients with BD in New Zealand [10,11,25,26,27]. This component of the study aimed to discuss the evidence about patterns of health service use and Māori patient experiences with focus group participants, and to draw on their expertise as healthcare professionals to develop guidelines for best practice for Māori with BD and generate strategies for change to address areas of unmet need.

## 2. Materials and Methods

This study involved focus groups conducted with health professionals responsible for the design and delivery of mental health services to Māori with BD and their whānau. Ethical approval was obtained from the Health and Disability Ethics Committee of New Zealand (ID: 16/STH/137). The CONSolIDated critERia (CONSIDER statement) for strengthening research involving Indigenous peoples were used to align the project with Indigenous research guidelines and priorities, and Kaupapa Māori methodology informed the choice and application of methods [27,28,29].

Three focus group sites were selected from different New Zealand regions to ensure inclusion of perspectives from urban and rural loci, and from health professionals delivering a range of mental health services representative of the types of services available to Māori patients across New Zealand. The MBDRP team established partnerships with focus group sites at the commencement of the project and maintained contact through each phase of the project. Relevant mental health service staff, with experience or responsibility for delivering mental health services to Māori with BD were identified prior to focus groups by liaising directly with established site contacts. Potential participants were emailed an information sheet inviting participation. Respondents received a consent form and focus group invitation. In-person focus groups were planned, with one completed in Christchurch on 16 March 2020. Due to COVID-19 disruptions, remaining focus groups were completed via video conferencing with Northland participants on 20 May 2020, and Hawke’s Bay participants on 3 September 2020. Written and verbal informed consent to participate was obtained before questioning. Four of the MBDRP team (CL, MCr, MCl, TH) facilitated focus groups in accordance with tikanga (Māori customary protocols) supported by kaumātua (Māori elder).

The three focus group questions were drawn from a synthesis of results from phases one and two of the MBDRP, detailed in Table 1. Questions were framed to elicit priority solutions from focus group participants to problems evident from phase one and two data, geared towards improving clinical, structural and organisational features of mental health service delivery for Māori patients with BD and their whānau. One month before the meeting, participants were provided with a synthesis of MBDRP findings, asked to consult with their sector colleagues, and prepare responses to focus group questions.

Nominal group technique (NGT) was selected for this study as a focus group method and analytic tool to assist with eliciting health service priorities [30,31]. Strengths in the technique included: the use of structure to prime busy health professional participants to formulate responses ahead of scheduled meetings; eliciting input from all participants to obtain diverse perspectives; and using discussion then ranking to prioritise action points in a timely and efficient way, which minimised the demand on participant’s time and required no additional analysis. At the end of the first focus group, priority responses to questions were collated, distributed by email to participants for verification and comment, and then finalised. A summary of priority responses from Christchurch was sent to participants in Northland and Hawke’s Bay for review, consideration, and consultation with sector colleagues ahead of the scheduled online focus groups. While the preparation for these groups was similar, the conduct of the group differed due to the challenges of conducting a focus group via teleconferencing.

For the in-person focus group, participant responses were summarised and transcribed onto a whiteboard by a member of the research team to enable group members to see how their comments were being captured and to ensure accuracy. Once participants had responded to each question, there was discussion about areas of overlap and commonalities to synthesise common proposed points of action. Condensed recommendations were checked with the group to ensure that this accurately captured key points without omitting important details. Participants cast a single vote for the condensed recommendation they considered to be the highest priority. Votes were then tallied to generate the rank order of priority action points. During teleconferencing focus groups, rather than voting and ranking, participants were invited instead to offer feedback and discussion about identified priority action points, which were captured digitally using the shared screen facility. Participants then offered an endorsement of the importance and priority ranking of identified responses to each of the focus group questions.

## 3. Results

A total of 22 key informants took part in three focus groups at three study sites. Sites were chosen for their differing range of mental health services, and their locations across New Zealand including urban and rural centres. Of the 22 focus group participants, all were staff involved in the design and delivery of mental health services for Māori patients with BD. Table 2 summarises the professional and other characteristics of focus group participants.

### 3.1. Responses to Question 1: What Ways Should You Be Working with Whānau to Reduce the Rate of Adverse Experiences for Māori with BD?

Participants identified four priority areas to reduce the rate of adverse experiences reported by Māori patients with BD and their whānau and improve clinical features of mental health service delivery. Action points are listed in order of priority in Table 3 and described in more detail below.

#### 3.1.1. Priority 1: Employ Tikanga-Based Engagement and Assessment Models

Participants expressed the need to adapt from current psychiatric models of engagement and assessment in mental health settings. The existing assessment model was identified as privileging organisational and professional priorities, with little alignment with Māori protocols. Participants recognised that engagement needed to be informed by an assessment process aligned with Māori protocols (incorporating traditional cultural practices, values and behaviours into the healthcare setting) that sought to resolve any differing views about risk. It was proposed that a revised process would reduce the potential escalation of conflict and distress that can arise through mismatched priorities between Māori patients with BD, their whānau, staff and services. 

There was strong agreement that alternate models must be structured to align with tikanga Māori, rather than adhering to the current psychiatric framework. Participants considered an adapted model was required to integrate tikanga and whakawhānaungatanga (the process of establishing a connection) through all clinical contacts, especially during initial engagement or transition points in care. There was recognition that an alternate model would require increased flexibility in services, including prioritising relationship building with Māori patients and whānau alongside data gathering, in the interests of providing equitable healthcare.

Several practical recommendations were made as to how this alternative model could be implemented, and the benefits of doing so were also explored. Wānanga (the process of meeting to share and develop knowledge) was considered one potential approach to initial engagement that could facilitate the identification of shared priorities between Māori patients with BD, whānau and mental health staff. Participants agreed that the use of wānanga would likely enhance subsequent assessment and care. Through incorporating tikanga, participants considered safety would be improved by ameliorating the power differential between consumers of healthcare and service providers. Utilising a tikanga-based model was also thought by participants to facilitate greater flexibility of service provision, by ensuring that clinical activities could occur safely across contexts, allowing mobility of care in or outside of mental health facilities matched to consumer needs. It was also recommended that the pressure to complete a comprehensive psychiatric assessment in the first encounter should be challenged and consideration given to staggering assessments. It was recognised that this model would require all staff to hold dual clinical/cultural competencies, and there was an acknowledgement that existing Māori health frameworks could be more widely utilised by services.

#### 3.1.2. Priority 2: Investment in the Māori Workforce

A lack of Māori workforce representation in the mental health sector, across all skills and specialities, was a key concern raised by focus group participants that needed to be prioritised and addressed to improve care for Māori with BD and their whānau. In addition to growing the number and range of Māori health professionals, further suggestions included increased funding for Māori employed in cultural support roles to work alongside clinicians and hiring Māori staff to facilitate service navigation between health systems.

#### 3.1.3. Priority 3: Resourcing Whānau

Participants discussed the importance of services investing time and resources to share health literacy information about BD with whānau from the time of first service contact onwards. Resourcing whānau was recognised as a pivotal step in the process towards achieving whānau ora (family wellbeing). Firstly, by supporting whānau to learn about BD, available treatments and the roles of services and providers. Secondly, by building meaningful partnerships between services and whānau for the benefit of Māori patients. The process of resourcing whānau was recognised as assisting health professionals to provide relevant information to enhance whānau knowledge of BD to support wellbeing (i.e., including the role of biopsychosociocultural factors in wellbeing), and to help whānau engage advocacy/support services when required to safely navigate the mental health system. Participants considered that resourcing could also extend to equipping whānau with evidence-based skills, such as training whānau in appropriate psychological therapies, to empower them to support their loved one with BD to remain well.

Participants considered additional resourcing should include conducting wānanga with whānau during periods of wellness to develop advance care plans to facilitate service access if required by Māori patients in future. It was envisaged that wānanga would also generate information to develop collaborative healthcare plans, based on shared understandings of patient, whānau and service priorities, and focused on wellbeing as opposed to the deficit-based medical model, which centres on symptomology.

#### 3.1.4. Priority 4: Investment in Professional Development and Service Evaluation

There was consensus amongst focus group participants about the need for all health professionals to be trained in both clinical and cultural competencies, with access to tools that facilitate engagement and ensure equitable healthcare. Participants recognised the need to support the workforce to access training to upskill, to have Māori staff available to allow flexibility around who leads communication with whānau and to embed evaluation into clinical encounters to continuously monitor and improve service responsiveness to Māori.

### 3.2. Responses to Question 2: Given the Likely Direction of the Move towards a Reduction in the Use of Community Treatment Orders, What Would Your Mental Health System Need to Do to Ensure That There Is Not a Loss of Support for Māori?

Participants identified three priority actions required to reduce the use of CTOs with Māori patients with BD and maintain essential support for patients and whānau. Action points are listed in priority order in Table 4 and are described in more detail below.

#### 3.2.1. Priority 1: Resourcing for Māori with Prior CTOs Matched to Early Intervention Service Level

Participants noted that in order to agree on a new resourcing model, it must be acknowledged that the over-representation of Māori placed under CTOs relative to non-Māori is the result of the cumulative impact of institutional racism. A transitional period in the reduction of CTOs would therefore be needed to allow for the development of a new model that ensures changes do not further disadvantage Māori. Participants agreed that adaptations needed to include the continued resourcing of effective care, such as ongoing free medication access, without the punitive aspects of CTOs. 

Participants considered a progressive model would need to retain any perceived benefits of CTOs from the perspective of Māori patients and their whānau, alongside an adaptation of existing processes to ensure Māori with BD receive equitable care from health services and providers. This could be achieved by creating a model that provides whānau with a cohesive support network of services, recognises whānau engagement as an essential component of providing care for Māori with BD, and requires communication with whānau from all personnel.

#### 3.2.2. Priority 2: Resourcing to Reduce the Impacts of Poverty and Adversity

Participants agreed that good quality healthcare begins with whānau having a clear understanding of best practices for Māori with BD, with health professionals playing an essential role in increasing the health literacy of whānau required to achieve hauora (wellbeing). Participants suggested effective resourcing was needed, based on early intervention models, to proactively identify and address barriers to accessing good quality healthcare (i.e., poverty, transport, ineffective engagement/communication, scheduling, etc.). A greater level of resourcing would be required to ensure that when barriers to best practice were identified, services were able to mitigate them, rather than being reliant on individual practitioner behaviour, advocacy or managerial discretion.

#### 3.2.3. Priority 3: Collaborative Whānau-Centred Multi-Agency/Multi-Service Shared Care Plans

Participants suggested that future health service models should at the outset, incorporate a treatment plan designed to equip Māori patients and their whānau with knowledge to live well with BD. In re-orienting to wellbeing, participants recognised the need for a revised whānau-centred model of care, where Māori patients and their whānau are recognised by services as stakeholders in healthcare planning, with their goals being an integral component of overall clinical management. It was also agreed that a shared care model should remain accessible across all services and agencies to facilitate timely and equitable treatment from different parts of the health system, including physical as well as mental healthcare providers, and services whose role is to mitigate the social determinants of health (i.e., housing, employment, education and justice). To be effective, it was acknowledged that shared care plans would also be needed to incorporate and be delivered in accordance with tikanga.

### 3.3. Responses to Question 3: What Structural Changes Could Improve Integration between Primary, Secondary and Other Services That Would Reduce Māori Admission Rates and Address Physical Comorbidities?

Participants identified three priority foci to improve the integration between primary, secondary, and other services and reduce rates of acute admission for Māori with BD as well as physical comorbidities. Action points are listed in priority order in Table 5 and described in more detail below.

#### 3.3.1. Priority 1: Resource a Comprehensive Shared Care Model

Participants agreed that integration between different healthcare services and systems would improve Māori health outcomes and reduce morbidity and mortality rates. They recognised that to achieve greater integration, existing communication issues between services must be addressed. This could be completed by implementing a holistic shared model of care that supports Māori patients with BD and their whānau to transition between services to ensure that all health priorities are completed. It was suggested that a mobile team could be established to maintain oversight of an overarching care plan for patients with BD, with responsibility for following up on Māori patient’s care needs regardless of where they are within the health system. It was recognised that a shared care model may require a broadening of responsibilities of the existing workforce (i.e., greater utilisation of Māori nurse practitioners), but that there would be benefits by removing siloes between services. As well as greater ease of information sharing, shared care benefits could also extend to individual services or providers being able to handover healthcare recommendations requiring continued monitoring and management if resources were invested in establishing mobile teams (i.e., adjusting medication to prevent over-sedation or improve sleep, treating diabetes, monitoring cardiovascular risk, etc.).

Participants considered that a comprehensive shared care plan was necessary to pool resources and reduce tensions and conflicts that exist under the current model where mental and physical healthcare priorities are managed separately rather than holistically. Shared care was seen as a necessary structural change to achieve health equity for Māori with BD, by minimising the risk of compounding existing systemic inequities to accessing quality healthcare, which already unfairly disadvantage Māori and people living with a mental illness.

#### 3.3.2. Priority 2: Flexible Mobile Healthcare Hubs

To reduce admission rates and address physical comorbidities for Māori, participants agreed that the existing design of healthcare delivery must be reconsidered. It was agreed that one point of contention continued to be “where health care is best delivered” for tangata whaiora (Māori person seeking wellness). Amongst participants, there were different ideas of where people should receive healthcare. Many participants thought that the existing model was not conducive to improving the lived experiences of some Māori patients and their whānau, and that the system needed to be more flexible in its response. An example given was that for some Māori patients with BD it may be a disservice to have their physical health needs managed only in a primary care setting—particularly since this may reduce access to physical healthcare during acute periods of illness. Participants thought future healthcare models should include the ability to discuss and adapt service delivery based on the changing needs of Māori patients and their whānau, including providing access to GPs within specialist mental healthcare settings.

Focus group participants also supported utilising social workers at front-end services as part of a flexible healthcare hub. It was considered that such an initiative would overcome the potential that primary care models may be biased towards a biomedical framework of healthcare. This may be particularly useful given the orientation of social work training, with the focus being much more ecological and aligned with broader concepts of hauora (holistic wellness) and the potential adverse impact of systems on Māori patients and whānau wellbeing.

#### 3.3.3. Priority 3: Access to Kai Ora (Healthy Food)

Focus group participants recognised the potential for kai (food) to play a greater role in achieving hauora for Māori with BD by reducing admission rates and physical comorbidities. There was discussion of the wellbeing potential of healthcare services incorporating access to healthy kai, and mātauranga Māori (Māori knowledge) practices like mahinga kai (traditional methods of producing, procuring and preparing food), as well as aligning with iwi (tribal) initiatives to promote wellbeing. It was agreed that future models should consider the provision of healthy kai within inpatient healthcare settings. Participants also recognised the potential for mahinga kai to be embedded in a sustainable way in community gardens or alongside mobile health hubs. Ideas included providing Māori patients with BD and their whānau with access to healthy kai, education and resources to grow, gather and prepare traditional kai, and opportunities for training or employment in mahinga kai. It was thought that structural changes such as these would enhance hauora not only by promoting healthy eating but also by supporting Māori patients and their whānau to achieve greater food security and increasing their connection to kai-centred Māori knowledges, practices and community networks.

## 4. Discussion

Focus group participants involved in the design and delivery of mental healthcare services utilised findings from the MBDRP to identify health service priorities and strategies necessary to improve mental health service delivery for Māori patients with BD and their whānau. At the clinical care level, service priorities identified by participants included the need to: transition from a psychiatric model of assessment and engagement to a tikanga-based model; invest in the development of a competent, diverse healthcare workforce; resource whānau to contribute to hauora alongside service input; and train staff to utilise hauora Māori frameworks and tools (such as the Hui Process and Meihana Model [32]) and embed evaluation into service provision to improve responsiveness to Māori. At the level of service structure, priorities identified by participants included the need to: increase resourcing for all Māori patients subject to compulsory treatment orders to equal early intervention input; introduce service elements to concurrently reduce the impact of social and Indigenous-specific determinants of hauora on Māori; and introduce shared care wellbeing-focused plans developed in collaboration with Māori patients and their whānau that remain accessible across different health services and systems. At the level of healthcare organisation, priorities identified by participants included the need to: resource a comprehensive shared care model to integrate physical and mental healthcare and remove siloes between services; create healthcare hubs that are flexible and mobile; and incorporate kai ora as an essential component of hauora.

This study extends beyond recent psychiatrically oriented clinical practice guidelines for the treatment of BD, by describing an approach to mental health service provision recommended by key informants to address existing inequities and meet Māori health needs [2]. Recommendations align with the intended outcomes of Māori health policy and proposed reform to improve the New Zealand mental health and addictions services while offering greater specificity about service delivery for Māori with BD and their whānau [17,20]. In addition, the need to devolve from psychiatry to a tikanga-based model of care is consistent with international systematic review findings that considered the variable prevalence of common mental health diagnoses in Indigenous samples was likely a reflection of limitations of psychiatric assessment and diagnostic frameworks perpetuating unmet needs [33]. The limitations of the dominant, biomedical model of health is well recognised internationally, with Indigenous and world health scholars alike identifying the health system as a determinant of health, and calling for reform to the organisation, structure and delivery of healthcare that aligns closely with these study findings [16,34,35].

Concordance was also high between the focus group priorities of mental health service providers to improve outcomes for Māori and recommended similar clinical, structural and organisational level changes to those identified by Māori patients and whānau in phase two of the broader MBDRP [11,25,26]. Focus group recommendations also aligned with integrated care research, which has shown that care coordination between primary, mental health and physical health providers greatly reduces costs and harmful care practices, improves outcomes for people with dual physical and mental healthcare needs, and has promise as a means to reduce systemic barriers and improve health outcomes for Indigenous peoples [36,37]. Although limited formal research has explored the contribution of Indigenous knowledges and practices to wellbeing, there is widespread acknowledgement of the “potential benefit” of holistic, culturally responsive healthcare, and cultural safety and competency in the design and delivery of equity-oriented clinical practice [38,39,40,41]. While more Indigenous-designed and led research is required in this area, there is widespread support for the necessity of service adaptations raised in this study, to provide care tailored to meet the needs of Māori patients and their whānau [38,39,40,41].

Strengths of this study include the use of a Kaupapa Māori Research design to present the broader MBDRP findings to focus group participants involved in the design and delivery of mental health services to establish priorities for system-level changes. There are also limitations of this study that need to be acknowledged. Firstly, the impact of COVID-19 restrictions necessitated a change in the modality from in-person to online for the last two focus groups, which required adaptation of the nominal group technique. Although the adaptation ensured focus group feedback was obtained from each study site, the challenge of facilitating discussions with participants remotely, who were also juggling essential roles in mental health service delivery during a global pandemic may have limited the generation of new or novel system-level changes. Despite this potential limitation, completing focus groups in this manner allowed study sites to be informed of broader findings, and for consensus to be reached about service changes required to improve mental health delivery for Māori patients with BD and their whānau from focus group participants from diverse disciplines, roles and perspectives. The authors also acknowledge that while the structured nominal group technique may have restricted participant commentary, the method also added value by ensuring every participant responded to each question, and by limiting the length of the focus groups to minimise disruptions to healthcare delivery during the study period.

## 5. Conclusions

The entrenched health inequities experienced internationally by Indigenous peoples, including Māori in New Zealand, reflect a violation of human rights, and the failure of successive governments to intervene [5,42,43,44,45]. The findings of this study provide further evidence of the need for healthcare reform in New Zealand, with recommendations centred on priorities to improve the responsiveness of the system for Māori with BD and their whānau [11,25,26]. Continued inaction when evidence of unmet needs is insurmountable is a hallmark of institutional racism [46]. Implementing evidence-based changes will require continued leadership from our mental health workforce, Māori and Tāngata Tiriti (non-Māori who are active in their roles as citizens and Treaty of Waitangi partners) alike, particularly while hard-fought gains towards health equity commitments in New Zealand remain so vulnerable to political influence [45,46,47]. The abandonment of Te Aka Whai Ora, the Māori Health Authority, is one recent example of sacrificing Māori health equity in favour of New Zealand politics [48,49].

## Figures and Tables

**Table 1 healthcare-12-00793-t001:** Synthesis of MBDRP findings informing focus group questions.

**Clinical Level Findings**	**Question 1**
Inpatient admissions were more common for Māori with BD relative to non-Māori and were almost always perceived as adverse experiences by both patients and whānau. This is despite the frequency of contacts in outpatient care being the same between Māori and non-Māori, and whānau being more commonly involved in outpatient care than non-Māori.	In what ways should you be working with whānau to reduce the rate of adverse experiences for Māori with BD?
**Structural Level Findings**	**Question 2**
The use of the Mental Health Act was more common for Māori with BD relative to non-Māori, in particular compulsory treatment orders (CTOs). Māori patients and their whānau spoke about the use of CTOs as at times providing a structure and frame for their treatment and facilitating their access to services and funded medications. Others spoke of the conflict between clinicians’ priorities for treatment planning and goals that did not align with Māori priorities for hauora (holistic wellbeing).	Given the likely direction of the move towards a reduction in the use of CTOs, what would your mental health system need to do to ensure that there is not a loss of support for Māori?
**Organisational Level Findings**	**Question 3**
National data identified that Māori with BD in secondary care were unlikely to have all of their health needs met relative to non-Māori and were dying more frequently. Māori patients and their whānau expressed aspirations for health systems to operate holistically (not in silos that separate primary, secondary and tertiary care) with the overarching goal of hauora in mind.	What structural changes could improve integration between primary, secondary and other services that would reduce admission rates and address physical comorbidities in Māori with bipolar disorder?

**Table 2 healthcare-12-00793-t002:** Characteristics of focus group participants.

Participant Details		N (22)	%
Gender	Female	15	68
	Male	7	32
Ethnicity	Māori	12	55
	Non-Māori	10	45
Discipline	Non-clinical	9	40
	Nursing	5	23
	Psychiatry	4	18
	Social work	2	9
	Psychology	1	5
	Addiction clinician	1	5
Role	Clinical	6	27
	Clinical or service management	4	18
	Māori health worker	2	9
	Cultural advisor	2	9
	Clinical director	2	9
	Executive director	2	9
	Funding and planning/training manager	2	9
	Consumer advisor	1	5
	Policy advisor	1	5
Location	Christchurch	11	50
	Hawke’s Bay	7	32
	Northland	4	18

**Table 3 healthcare-12-00793-t003:** Clinical priorities.

Priority Rank	Action Point
1	Employ different engagement and assessment models based on tikanga
2	Investment in the Māori workforce
3	Resource the whānau to improve their understanding of BD to enable them to contribute to hauora (wellbeing)
4	Investment in professional development and service evaluation

**Table 4 healthcare-12-00793-t004:** Structural priorities.

Priority Rank	Action Point
1	Increase resourcing for Māori with prior CTOs to a level of service equivalent to early intervention service resourcing
2	Introduce elements of service provision that reduce the impact of poverty and adversity
3	Develop shared care plans for BD collaboratively with whānau alongside other health services and providers

**Table 5 healthcare-12-00793-t005:** Organisational priorities.

Priority Rank	Action Point
1	Resource a comprehensive shared care health model
2	Create flexible mobile healthcare hubs
3	Provide access to kai ora (healthy food)

## Data Availability

The original contributions presented in the study are included in the article. The raw data are not publicly available due to the fact that this was not subject to informed consent, and is not consistent with the CONSIDER statement adopted by the authors to strengthen the reporting of Indigenous health research [27,28].

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
