# Peer review of "The Right to Equal Health: Best Practice Priorities for Māori with Bipolar Disorder from Staff Focus Groups"

_healthcare, 2024, doi:10.3390/healthcare12070793_

Round 1
Reviewer 1 Report
Comments and Suggestions for Authors
The paper looks very interesting and informative.
Though exciting, it is sometimes confusing because it does not set a good tone about the cultural background.
Introduction
Readers from other countries are not well-oriented with the culture of New Zealand.
Briefly explain the mental health model in New Zealand.
Please provide a brief background about the Maori and their health models.
Lines
156- please expand on Maori protocols
291- spelling typo - ( 8ecognized )
335- please describe the meaning of Hauora.
Limitations
Sample size
Also, I feel that nonclinical participants' opinions could be confounding because of their knowledge about bipolar.
The discussion was vague about Hauora Maori frameworks and tools.
Comments on the Quality of English LanguageIt could be improved.
Author Response
Please see the attachment. Response to both reviewers included for ease of reference.

Reviewer 2 Report
Comments and Suggestions for Authors
This study aimed to share evidence about patterns of health service use and Māori patient experiences with participants to develop guidelines for best practice for Māori with BD to address areas of unmet need. There are several issues that should be considered.
1. It is challenging to draw firm conclusions because of the qualitative approach and the limited sample of individuals that were included.
2. This investigation was carried out in a single mental health facility. The results might not apply to other mental health treatments. There exist regional disparities in New Zealand with regards to the percentage of Māori people and the percentage of people living in socioeconomic deprivation. Since Māori identity is not uniform, the results might only reflect the opinions of people who reside in a certain region.
3. Recruiting Māori patients and whānau through health care providers may have limited participation to people with positive service experiences.
4. The conclusion should not include any references or any new information.
5. The ability to handle only one issue at a time and the requirement for a particular level of conformance from all participants make the Nominal Group Technique (NGT) rigid.
Author Response
Please see the attachment. I have included responses to both reviewers for ease of reference.

Round 2
Reviewer 2 Report
Comments and Suggestions for Authors
The authors address my comments